# A Perspective Distortion Correction Method for Planar Imaging Based on Homography Mapping

**DOI:** 10.3390/s25061891

**Published:** 2025-03-18

**Authors:** Chen Wang, Yabin Ding, Kai Cui, Jianhui Li, Qingpo Xu, Jiangping Mei

**Affiliations:** Key Laboratory of Mechanism Theory and Equipment Design of State Ministry of Education, Tianjin University, Tianjin 300350, China; chenw1007@tju.edu.cn (C.W.); iscuik@tju.edu.cn (K.C.); isjhli@163.com (J.L.); qingpoxu@tju.edu.cn (Q.X.); ppm@tju.edu.cn (J.M.)

**Keywords:** monocular vision, precision measurement, pixel equivalent, perspective distortion, homography mapping

## Abstract

In monocular vision measurement, a barrier to implementation is the perspective distortion caused by manufacturing errors in the imaging chip and non-parallelism between the measurement plane and its image, which seriously affects the accuracy of pixel equivalent and measurement results. This paper proposed a perspective distortion correction method for planar imaging based on homography mapping. Factors causing perspective distortion from the camera’s intrinsic and extrinsic parameters were analyzed, followed by constructing a perspective transformation model. Then, a corrected imaging plane was constructed, and the model was further calibrated by utilizing the homography between the measurement plane, the actual imaging plane, and the corrected imaging plane. The nonlinear and perspective distortions were simultaneously corrected by transforming the original image to the corrected imaging plane. The experiment measuring the radius, length, angle, and area of a designed pattern shows that the root mean square errors will be 0.016 mm, 0.052 mm, 0.16°, and 0.68 mm^2^, and the standard deviations will be 0.016 mm, 0.045 mm, 0.033° and 0.65 mm^2^, respectively. The proposed method can effectively solve the problem of high-precision planar measurement under perspective distortion.

## 1. Introduction

Vision measurement technology has played essential roles in several industrial sectors, particularly in intelligent manufacturing systems [1,2]. This technology leverages images for non-contact measurement, which circumvents the limitations of contact measurement with traditional measuring tools. It offers several advantages, including high precision, enhanced efficiency, non-scratch workpieces, and real-time tracking of the measurement results.

When measuring 2D parameters such as radius, length, angle, and area of a target, a monocular vision system is typically employed. In this system, the target is placed on the measurement plane, and its image is captured by a camera installed vertically above the plane. The pixel equivalent method is then applied to measure the target’s 2D parameters through calibrating the actual physical size corresponding to a single pixel [3,4,5,6]. The maximum accuracy that the system can achieve is an important factor limiting its engineering application. To improve the accuracy, two primary factors must be considered. First, the image edge detection algorithm should extract the target’s edge accurately. Second, the captured image has minimal perspective distortion, which requires a finely manufactured imaging chip and a perfect parallelism between the measurement plane and its image [7,8,9]. If the manufacturing or the parallelism is poor, the measurement plane will produce perspective distortion in imaging, leading to an inaccurate pixel equivalent and ultimately affecting the measurement accuracy.

The existing solutions for the perspective distortion problem can be divided into two categories: hardware and algorithm.

At the hardware level, the perspective distortion can be minimized by carefully adjusting the camera’s position relative to the measurement plane [10]. However, the captured images may still have a little perspective distortion. In some application scenes, the camera may need to be tilted to capture the measurement plane. Another approach involves using a high-cost telecentric lens instead of a standard optical lens [11,12]. However, the telecentric lens must cover the entire measurement plane during application, making it unapplicable for the measurement with a large field of view. In addition, although the telecentric lens provides consistent magnification in depth direction, it also leads to oblique projection. As depicted in Figure 1, if a checkboard is tilted relative to the camera, the aspect ratio of the board will be altered in telecentric lens imaging. The third approach is adding additional sensors [13,14,15]. Liu S, Ge Y, Wang S et al. used a monocular camera with structured light to measure the center distance of planar holes, with a measurement accuracy of 0.1 mm [13]. Chen L, Zhong G, Han Z et al. used a binocular camera to measure the size of planar rectangular workpieces, achieving a measurement accuracy of 0.02 mm [14]. Although this approach effectively avoids perspective errors, adding additional sensors undoubtedly increases both the difficulty of calibration and the overall cost. Consequently, solving the perspective distortion problem from the hardware level either increases costs or cannot be fully solved.

At the algorithmic level, one approach is using the coordinate transformation method [16,17,18]. The target’s world coordinates are computed through the calibrated camera imaging model in the method. Zhang X and Yin H measured the cable cross-sectional radius by aligning a checkboard with the cable cross-section during measurement, with a measurement accuracy of 0.768 mm [16]. Similarly, Miao J, Tan Q, Liu S et al. used a checkboard aligned to the gear end face to calibrate the camera imaging model and measured the gear’s pitch error within 0.06 mm [17]. Although this method can effectively avoid the errors caused by perspective distortion, it is inefficient as it requires calculating world coordinates point by point. Consequently, it is only suitable for measuring simple and regular shapes, not complex and irregular shapes, especially the area of irregular planar shapes. Another approach is correcting perspective distortion by calibrating the conventional perspective transformation model, including calibration based on vanishing points or control point pairs. The former calibrates the model by detecting the vanishing points in images [19,20,21]. Lin J and Peng J corrected the perspective distortion of railroad track images using vanishing points and further realized the tracks’ visual detection [20]. However, the vanishing points are susceptible to camera nonlinear distortion and the existence of parallel lines, resulting in lower correction accuracy. The method is generally used for perspective correction of buildings, roads, and text. The latter calibrates the model by setting the control point pairs in images [22,23]. Wang Q, Zhou Q, Jing G et al. corrected the perspective distortion of circular saw core images by utilizing a checkboard to set control point pairs [22]. The precise position of the circular saw core was then obtained through calibrated pixel equivalent. However, the ideal control points corresponding to the distorted points in this approach are set manually, which can easily create a zoom effect on the whole image, affecting the pixel equivalent accuracy while causing the corrected image to lose the distance scale of the measurement plane relative to the camera. Additionally, the correction accuracy relies heavily on the precision of the manually set ideal control points, i.e., the corrected image will still have a large perspective distortion if the ideal control points are set improperly. The last approach is correcting perspective distortion by deep learning techniques [24,25]. However, this method requires extensive accurate sample for model training and necessitates retraining when the target changes. Consequently, its training cost significantly exceeds that of conventional methods. Moreover, unlike conventional approaches, its model parameters lack explicit physical meaning, which compromises the interpretability of the system.

The foregoing discussion shows that monocular vision measurement technology lacks a low-cost, automated, and highly accurate method to correct planar perspective distortion. If this method can be proposed, the accuracy of the pixel equivalent method can be further improved, and its application can be further expanded. To achieve this purpose, this paper proposed a perspective distortion correction method for planar imaging based on homography mapping and built an experimental platform to verify its effectiveness. First, factors causing perspective distortion of the measurement plane were analyzed from the camera’s intrinsic and extrinsic parameters, and a perspective transformation model of the image was constructed. A corrected imaging plane was then constructed based on the adjustment of the parameters. The established model was then calibrated by utilizing the homography between the measurement plane, the actual imaging plane, and the corrected imaging plane. The nonlinear and perspective distortions in the original image were simultaneously corrected by transforming the original image to the corrected imaging plane. Finally, the proposed method was verified to effectively correct perspective distortion in planar imaging and maintain higher measurement accuracy and stability compared to existing methods.

The remainder of this paper is organized as follows. The causes of perspective distortion are analyzed in Section 2. The distortion correction method is proposed in Section 3. Experiments evaluating the methods’ performance are presented in Section 4 before conclusions are drawn in Section 5.

## 2. Perspective Distortion Analysis

Analyzing the causes of perspective distortion is a prerequisite for its correction. This section analyzes the causes from the camera’s intrinsic and extrinsic parameters according to the pinhole imaging model.

### 2.1. Analysis from Extrinsic Parameters

When it comes to perspective distortion in planar imaging, the first factor that comes to mind is that the camera’s optical axis is not exactly perpendicular to the measurement plane resulting in the non-parallelism between the measurement plane and the camera’s imaging plane. This non-parallelism can be reflected in the transformation of the world coordinate system $O_{w}-X_{w}Y_{w}Z_{w}$ to the camera coordinate system $O_{c}-X_{c}Y_{c}Z_{c}$ according to the pinhole imaging model. Define $O_{w}-X_{w}Y_{w}Z_{w}$ on the measurement plane, where $Z_{w}=0$ coincides with the measurement plane, then the transformation can be presented as:(1)$$\left[\begin{array}{c} X_{c}\\ Y_{c}\\ Z_{c}\\ 1 \end{array}\right]=\left[\begin{array}{l} \begin{array}{cc} R & \;\;t \end{array}\\ \begin{array}{cc} \vec{0} & 1 \end{array} \end{array}\right]\left[\begin{array}{c} X_{w}\\ Y_{w}\\ 0\\ 1 \end{array}\right]$$
where $t={\left[\begin{array}{ccc} t_{1} & t_{2} & t_{3} \end{array}\right]}^{T}$ is the 3 × 1 translation vector, $R=[\begin{array}{ccc} r_{1} & r_{2} & r_{3} \end{array}]$ is the 3 × 3 rotation matrix.

From Equation (1), if the measurement plane is parallel to the imaging plane, the *Z_c_* values of the points $\left(X_{w},Y_{w}\right)$ on the measurement plane are equal, and at this point, **R** is an ideal matrix. When the two are not parallel, **R** becomes non-ideal, so **R** can be used to quantitatively assess the parallelism between the planes. However, the parameters in **R** are numerous and unintuitive; thus, **R** is further decomposed into Euler angles. Assuming that $O_{w}-X_{w}Y_{w}Z_{w}$ first rotates the Euler angle *γ* around the *X_c_* axis, then rotates the Euler angle *β* around the *Y_c_* axis, and finally rotates the Euler angle *α* around the *Z_c_* axis, then **R** can be decomposed as:(2)$$R=\left[\begin{array}{ccc} \cos\alpha & -\sin\alpha & 0\\ \sin\alpha & \cos\alpha & 0\\ 0 & 0 & 1 \end{array}\right]\left[\begin{array}{ccc} \cos\beta & 0 & \sin\beta \\ 0 & 1 & 0\\ -\sin\beta & 0 & \cos\beta \end{array}\right]\left[\begin{array}{ccc} 1 & 0 & 0\\ 0 & \cos\gamma & -\sin\gamma \\ 0 & \sin\gamma & \cos\gamma \end{array}\right]$$

According to Equations (1) and (2), the *Z_c_* value of each $\left(X_{w},Y_{w}\right)$ can be computed as:(3)$$Z_{c}=t_{3}-\sin\beta X_{w}+\cos\beta \sin\gamma Y_{w}$$

From Equation (3), the *Z_c_* value of each $\left(X_{w},Y_{w}\right)$ is equal to *t*_3_ when *β* and *γ* are equal to zero, at which point the measurement plane is completely parallel to the imaging plane. Further, the impact of the three Euler angles on plane imaging is shown in Figure 2.

Through the above analysis, the Euler angles *β* and *γ* in extrinsic parameters cause perspective distortion in planar imaging.

### 2.2. Analysis from Intrinsic Parameters

Even if the measurement plane is completely parallel to the imaging plane, the perspective distortion caused by manufacturing errors in camera imaging chips cannot be ignored. The manufacturing errors can be reflected in the camera’s intrinsic matrix **K**:(4)$$K=\left[\begin{array}{ccc} f_{x} & s & u_{0}\\ 0 & f_{y} & v_{0}\\ 0 & 0 & 1 \end{array}\right]$$
where $\left(u_{0},v_{0}\right)$ represents the pixel coordinate of the principal point, s denotes the oblique factor of the imaging chip, $f_{x}$ and $f_{y}$ represent the equivalent focal lengths in x and y directions, respectively.

Meanwhile, $f_{x}$ and $f_{y}$ can be computed as:(5)$$\left\{\begin{array}{l} f_{x}=F/p_{x}\\ f_{y}=F/p_{y} \end{array}\right.$$
where *F* represents the physical focal length of the camera, $p_{x}$ and $p_{y}$ denote the length and height of the pixel unit, respectively.

As depicted in Figure 3, the perspective distortion caused by **K** mainly has two factors. One is that the smallest unit of the imaging chip is not a perfect square, i.e., $p_{x}$ is not equal to $p_{y}$. This makes the camera’s magnification in x, and y directions inconsistent, causing $f_{x}$ and $f_{y}$ to be unequal. Another is that the imaging chip itself is not exactly perpendicular, existing a dip angle *θ*, resulting in $s=f_{x}\cot\theta$.

## 3. Proposed Method

### 3.1. Model Construction

To correct the perspective distortion, a perspective transformation model is constructed to perform perspective transformation on original images:(6)$$\lambda_{1}\left[\begin{array}{c} u^{\prime }\\ v^{\prime }\\ 1 \end{array}\right]=T\left[\begin{array}{c} u\\ v\\ 1 \end{array}\right]$$
where $\lambda_{1}$ represents the scale factor, (u,v) denotes the pixel coordinate of the actual perspective distorted point in the original image, $\left(u^{\prime },v^{\prime }\right)$ denotes the pixel coordinate of the undistorted point in the corrected image, and **T** is a 3 × 3 perspective transformation matrix with 8 degrees of freedom.

### 3.2. Model Calibration

In traditional calibration methods for **T**, the detection of (u,v) from the original images often neglects nonlinear distortion. Additionally, the corresponding $\left(u^{\prime },v^{\prime }\right)$ need to be manually set based on empirical knowledge, which results in low calibration accuracy of **T** and often introduces a zoom effect on the whole image [22,23]. To address these limitations, the study constructs a nonlinear distortion model and subsequently calibrates **T** by leveraging the homography relationship between planes.

#### 3.2.1. Calibration of the Actual Homography Matrix

The actual homography matrix reflects the homography relationship between the measurement plane and the actual imaging plane. As depicted in Figure 4, a pre-calibrated monocular camera is used to capture the original image of a checkboard placed on the measurement plane. Then, the world coordinate system $O_{w}-X_{w}Y_{w}Z_{w}$ is established on the checkboard with its origin $O_{w}$ at the upper-left corner and its plane $Z_{w}=0$ coinciding with the measurement plane. According to the pinhole imaging model, the mapping relationship from the checkboard world points $\left(X_{w},Y_{w}\right)$ to its corresponding (u,v) on the actual imaging plane can be represented as [26]:(7)$$\lambda_{2}\left[\begin{array}{l} u\\ v\\ 1 \end{array}\right]=K\left[\begin{array}{ccc} r_{1} & r_{2} & t \end{array}\right]\left[\begin{array}{c} X_{w}\\ Y_{w}\\ 1 \end{array}\right]$$
where $\lambda_{2}$ represents the scale factor.

According to the theory of projective transformation, there exists a homography mapping relationship between corresponding coordinate points on the measurement plane and the actual imaging plane. This relationship can be mathematically constructed as a 3 × 3 matrix **H_real_**:(8)$$\lambda_{2}\left[\begin{array}{l} u\\ v\\ 1 \end{array}\right]=H_{real}\left[\begin{array}{c} X_{w}\\ Y_{w}\\ 1 \end{array}\right]$$

Observing Equation (7), both **K** and $\left[\begin{array}{ccc} r_{1} & r_{2} & t \end{array}\right]$ are 3 × 3 matrices, which proves that $K[\begin{array}{ccc} r_{1} & r_{2} & t \end{array}]$ is also a 3 × 3 matrix and reflects the homography mapping relationship between $\left(X_{w},Y_{w}\right)$ and (u,v). So **H_real_** can be further expressed as:(9)$$H_{real}=\left[\begin{array}{ccc} h_{1} & h_{2} & h_{3} \end{array}\right]=\left[\begin{array}{ccc} h_{11} & h_{12} & h_{13}\\ h_{21} & h_{22} & h_{23}\\ h_{31} & h_{32} & 1 \end{array}\right]=K\left[\begin{array}{ccc} r_{1} & r_{2} & t \end{array}\right]$$

The actual homography matrix **H_real_** has 8 degrees of freedom. Hence, it can be calibrated using more than 4 pairs of $\left(X_{w},Y_{w}\right)$ and its corresponding (u,v).

However, nonlinear distortion is inevitable due to manufacturing errors in the camera’s optical system [26]. The coordinates of the checkboard corners $\left(u_{d},v_{d}\right)$ detected from the original image need to correct nonlinear distortion to obtain (u,v). For the pre-calibrated camera, its radial distortion parameters are named as $k_{1},k_{2}$, its tangential distortion parameters are named as $p_{1},p_{2}$, then the distortion model can be expressed as:(10)$$\left[\begin{array}{c} x_{d}\\ y_{d} \end{array}\right]=(1+k_{1}r^{2}+k_{2}r^{4})\left[\begin{array}{c} x\\ y \end{array}\right]+\left[\begin{array}{c} 2p_{1}x+p_{2}(r^{2}+2x^{2})\\ p_{1}(r^{2}+2y^{2})+2p_{2}y \end{array}\right]$$
where $r=\sqrt{x^{2}+y^{2}}$ and $\left(x_{d},y_{d}\right)$ represent the coordinate containing nonlinear distortion on the normalized plane, and its transformation with $\left(u_{d},v_{d}\right)$ is as follows:(11)$$\left\{\begin{array}{c} x_{d}=\left(u_{d}-u_{0}-s\cdot y_{d}\right)/f_{x}\\ y_{d}=\left(v_{d}-v_{0}\right)/f_{y} \end{array}\right.$$
where (x,y) represents the coordinate without nonlinear distortion on the normalized plane, and its transformation with (u,v) is as follows:(12)$$\left\{\begin{array}{c} x=\left(u-u_{0}-s\cdot y\right)/f_{x}\\ y=\left(v-v_{0}\right)/f_{y} \end{array}\right.$$

By substituting $\left(u_{d},v_{d}\right)$ into Equations (10)–(12), (u,v) can be obtained, and further **H_real_** can be calibrated.

#### 3.2.2. Generation of the Corrected Homography Matrix

Based on the analysis in Section 2, a virtual camera is created by adjusting the actual camera’s intrinsic and extrinsic parameters. The imaging plane of the virtual camera is defined as the corrected imaging plane, which is completely parallel to the measurement plane.

Intrinsic parameter adjustment:

Intrinsic parameters adjustment is to obtain the virtual camera’s intrinsic matrix **K′** based on **K**. The oblique factor of **K′** is set to zero. *F* remains the same but the single pixel unit’s length and height are set to $\sqrt{p_{x}p_{y}}$. Combine Equation (5), the equivalent focal lengths $f_{x}^{\prime }$, $f_{y}^{\prime }$, of **K′** can be computed as:(13)$$f_{x}^{\prime }=f_{y}^{\prime }=F/\sqrt{p_{x}p_{y}}=\sqrt{f_{x}f_{y}}$$

Thus, **K′** can be expressed as:(14)$$K\prime =\left[\begin{array}{ccc} \sqrt{f_{x}f_{y}} & 0 & u_{0}\\ 0 & \sqrt{f_{x}f_{y}} & v_{0}\\ 0 & 0 & 1 \end{array}\right]$$

Extrinsic parameter adjustment:

According to Equations (7) and (8), **R** and **t** can be computed by matrix decomposition:(15)$$\left\{\begin{array}{l} r_{1}=(1/\lambda_{3})K^{-1}h_{1}\\ r_{2}=(1/\lambda_{3})K^{-1}h_{2}\\ r_{3}=r_{1}\times r_{2}\\ t=(1/\lambda_{3})K^{-1}h_{3} \end{array}\right.$$
where $\lambda_{3}$ represents the scale factor.

Extrinsic parameters adjustment refers to adjusting **R** by decomposing it into α,β,γ, then only α is preserved, β and γ are set to zero. The adjusted rotation matrix **R′** can be expressed as:(16)$$R^{\prime }=[\begin{array}{ccc} r_{1}^{\prime } & r_{2}^{\prime } & r_{3}^{\prime } \end{array}]=\left[\begin{array}{ccc} \cos\alpha & -\sin\alpha & 0\\ \sin\alpha & \cos\alpha & 0\\ 0 & 0 & 1 \end{array}\right]$$

At this point, the virtual camera can be created by setting its intrinsic and extrinsic parameters to **K′**, **R′** and **t**. Substituting these parameters into Equation (7), The corrected homography matrix **H_rect_** from $\left(X_{w},Y_{w}\right)$ to its corresponding $\left(u^{\prime },v^{\prime }\right)$ on the corrected imaging plane can be generated as:(17)$$\lambda_{4}\left[\begin{array}{l} u^{\prime }\\ v^{\prime }\\ 1 \end{array}\right]=K^{\prime }[\begin{array}{ccc} r_{1}^{\prime } & r_{2}^{\prime } & t \end{array}]\left[\begin{array}{c} X_{w}\\ Y_{w}\\ 1 \end{array}\right]=H_{rect}\left[\begin{array}{c} X_{w}\\ Y_{w}\\ 1 \end{array}\right]$$
where $\lambda_{4}$ presents the scale factor.

#### 3.2.3. Calibration

From Equation (8), we can obtain the following:(18)$$\left[\begin{array}{c} X_{w}\\ Y_{w}\\ 1 \end{array}\right]=\lambda_{2}H_{real}^{-1}\left[\begin{array}{l} u\\ v\\ 1 \end{array}\right]$$

Then, substitute Equation (18) into Equation (17):(19)$$\lambda_{4}\left[\begin{array}{l} u\prime \\ v\prime \\ 1 \end{array}\right]=\lambda_{2}H_{rect}H_{real}^{-1}\left[\begin{array}{l} u\\ v\\ 1 \end{array}\right]$$

Combine Equation (19) with Equation (6):(20)$$T=H_{rect}H_{real}^{-1}$$

At this point, the calibration of **T** is realized. The whole homography relationship is shown in Figure 5. After correcting the nonlinear distortion, the perspective distorted image on the actual imaging plane can be transformed into an undistorted image on the corrected imaging plane through Equation (6).

The imaging of the corrected image conforms to the pinhole imaging principle of the virtual camera. We can find the object distance for each $\left(X_{w},Y_{w}\right)$ to the virtual camera is $t_{3}$ through Equation (3). Meanwhile, the virtual camera’s equivalent focal lengths in x, and y directions are equal to $\sqrt{f_{x}f_{y}}$, implying that the pixel equivalents *M* in x, and y directions are equal and can be computed as:(21)$$M=t_{3}/\sqrt{f_{x}f_{y}}$$

#### 3.2.4. Method to Improve Calibration Accuracy

The transformation relationship of the four types of coordinate points in this study is depicted in Figure 6. One readily derives, **T** is determined by **H_real_** and **H_rect_**. Meanwhile, **H_rect_** is computed with **K** and **H_real_**. Thus, the accuracy of **T** depends on the calibration accuracy of the camera and **H_real_**. Assume that the camera has a high calibration accuracy, to improve the accuracy of **T**, **H_real_** is calibrated as follows:

First, rearrange Equations (8) and (9):(22)$$\left\{\begin{array}{c} u=\left(h_{11}X_{w}+h_{12}Y_{w}+h_{13}\right)/\left(h_{31}X_{w}+h_{32}Y_{w}+1\right)\\ v=\left(h_{21}X_{w}+h_{22}Y_{w}+h_{23}\right)/\left(h_{31}X_{w}+h_{32}Y_{w}+1\right) \end{array}\right.$$

Further, transforming Equation (22), we can obtain the following:(23)$$\left[\begin{array}{c} \begin{array}{cccccccc} X_{w} & Y_{w} & 1 & 0 & 0 & 0 & -X_{w}u & -Y_{w}u \end{array}\\ \begin{array}{cccccccc} 0 & 0 & 0 & X_{w} & Y_{w} & 1 & -X_{w}v & -Y_{w}v \end{array} \end{array}\right]\eta^{T}=\left[\begin{array}{c} u\\ v \end{array}\right]$$
where $\eta =[\begin{array}{cccccccc} h_{11} & h_{12} & h_{13} & h_{21} & h_{22} & h_{23} & h_{31} & h_{32} \end{array}]$.

A pair of $\left(X_{w},Y_{w}\right)$ and its corresponding (u,v) can get two linear equations, which means **H_real_** can be calibrated by 4 pairs of the matching points. But to minimize the impact of image noise and errors in checkerboard corner extraction, this study selects a maximal number of checkboard corners and utilizes the least squares method to calibrate **H_real_**. Meanwhile, the checkboard used for calibration should have better production accuracy and imaging quality to ensure accuracy.

#### 3.2.5. Algorithm Design

In traditional methods, perspective distortion correction either neglects nonlinear distortion entirely or employs a two-stage image interpolation process, where nonlinear distortion correction is performed prior to perspective correction [19,20,21,22,23]. Both approaches inevitably lead to a degradation in the accuracy of the corrected images.

To ensure accuracy, this study integrates **T** with the nonlinear distortion model, enabling simultaneous correction of both perspective and nonlinear distortions through a single image interpolation process. The proposed distortion correction algorithm primarily consists of forward mapping and backward mapping, as depicted in Figure 7.

Initially, the forward mapping is employed to determine the size of the corrected image. Assume that the width and height of the original image be $w_{d}$ and $h_{d}$, respectively. The four corner coordinates of the original image can be expressed as (0,0), $\left(0,h_{d}\right)$, $\left(w_{d},0\right)$, $\left(w_{d},h_{d}\right)$, then substitute them as $\left(u_{d},v_{d}\right)$ into Equations (10)–(12) to obtain four intermediate coordinates corresponding to (u,v). Afterward, the four intermediate coordinates are substituted into Equation (6) as (u,v) to obtain four new coordinates corresponding to (u′,v′): $\left(x_{1},y_{1}\right)$, $\left(x_{2},y_{2}\right)$, $\left(x_{3},y_{3}\right)$, $\left(x_{4},y_{4}\right)$. Finally, the corrected image’s width $w^{\prime }$ and height $h^{\prime }$ can be computed as:(24)$$\left\{\begin{array}{c} w\prime =ceil\left(\max(x_{1},x_{2},x_{3},x_{4})-\min(x_{1},x_{2},x_{3},x_{4})+1\right)\\ h\prime =ceil\left(\max(y_{1},y_{2},y_{3},y_{4})-\min(y_{1},y_{2},y_{3},y_{4})+1\right) \end{array}\right.$$
where *ceil*() is the roundup function.

Furthermore, the reverse mapping is used to obtain the gray value of the corrected image. Begin by creating a blank image of the corrected image with a width of $w^{\prime }$ and a height of $h^{\prime }$. For each pixel (u′,v′) in the blank image, compute its corresponding (u,v) using Equation (6). Then, apply Equations (10)–(12) to compute $\left(u_{d},v_{d}\right)$ corresponding to (u,v). Afterward, the gray value at $\left(u_{d},v_{d}\right)$ is obtained by the bicubic interpolation method and is assigned to (u′,v′). After traversing all (u′,v′), the corrected image can be output.

## 4. Experiment

### 4.1. Experiment Design

To verify the effectiveness of the proposed method, the designed visual measurement platform is shown in Figure 8, which includes a PC, industrial camera, machine vision platform, ring light, adjustment block, and test board. The camera consists of an MER-500-14U3M CMOS (Daheng Imaging, Beijing, China) and a M1214-MP2 12 mm FA fixed focal lens (Computar, Tokyo, Japan). The specific physical parameters of the camera are shown in Table 1. Using the proposed method, the image processing and measurement experiments are implemented in MATLAB2020a on the PC running a 64-bit Windows 10 system with 2.5 GHz CPU and 8 GB RAM.

The test board is made of a 1 mm thick flat matte ceramic plate. Its pattern is shown in Figure 9, and the overall production accuracy of the pattern is ±1 µm. Each checkerboard square in the pattern is 5 mm in length, which is used to calibrate the perspective transformation model. Additionally, to avoid measurement errors caused by misalignment between the measurement plane and the calibration plane [8], four standard test patterns, coplanar with the checkboard pattern, are set on the test board to test the measurement performance of the proposed method. The circle pattern has a radius of 16 mm (labeled R16) and is used to evaluate radius measurement performance. The rectangle pattern has a length of 36 mm and a height of 22 mm (labeled L36 and H22, respectively) and is used to evaluate length measurement performance. The right triangle pattern has two acute angles of 37° and 53° (labeled D37 and D53, respectively) and is used to evaluate angle measurement performance. The ellipse pattern is viewed as an irregular planar shape with an area of 622.035 mm^2^ (labeled Area) and is used to evaluate area measurement performance.

Step 1: Camera calibration. A camera calibration board is placed on the platform, and multiple images of the board are captured by changing its poses relative to the camera. Then, the camera is calibrated by Zhang’s method.

Step 2: Image acquisition. The test board is placed on the platform. The position of the adjustment block beneath the test board is randomly adjusted to obtain images of the test board with different degrees of perspective distortion. To verify the repeatability of the proposed method, fifty original images of the test board in various poses are acquired. Specifically, half are captured during the day and the other half at night. Note that the test board should remain within the camera’s field of view and depth of field throughout the entire capture process. Meanwhile, the test board is well illuminated during the experiments.

Step 3: Image processing. The pixel coordinates $\left(u_{d},v_{d}\right)$ in the original images are obtained by checkboard corner detection. The **T** of each original image is then calibrated by the proposed method. Further, the corrected images are obtained by the designed correction algorithm.

Step 4: Experimental verification. To demonstrate the effectiveness of the method, four experiments are set up: **Exp.1** is to verify its accuracy by calculating the reprojection error of **T**; **Exp.2** is to evaluate the residual perspective distortion in the corrected images by computing their extrinsic parameters; **Exp.3** is to compare the measurement accuracy with the existing methods by measuring R16, L36, H22, D37, D53, and Area; **Exp.4** is to investigate the impact of camera calibration errors on the proposed method.

### 4.2. Results and Evaluations

Using the Camera Calibrator App in MATLAB2020a, the camera intrinsic parameters were calibrated through Zhang’s method, and the calibration results are shown in Table 2. The overall reprojection error of camera calibration results obtained from the App is 0.055 pixels.

The computed **K′** is shown as follows:(25)$$K^{\prime }=\left[\begin{array}{ccc} 5497.138 & 0 & 1292.928\\ 0 & 5497.138 & 960.322\\ 0 & 0 & 1 \end{array}\right]$$

Under the experimental conditions, including the specific computer hardware and camera resolution employed, the designed algorithm achieves an average correction time of 1.03 s per image.

To conduct **Exp.1** and **Exp.2**, nine images with different degrees of perspective distortion were selected from the fifty original images. Their extrinsic parameters, *α*, *β*, *γ* and *t*_3_, are shown in Table 3.

Of the nine selected images, Pose 4 and Pose 8, which have severe perspective distortion according to *β* and *γ*, were selected as examples. The comparison between their original and corrected images is shown in Figure 10.

**Exp.1**: Compute the reprojection error to verify the accuracy of **T**. After calibrating **T** and obtaining the corrected images corresponding to the selected nine original images, the pixel coordinates $\left(u^{\prime \prime },v^{\prime \prime }\right)$ of the checkboard corners in each corrected image were re-detected. Their corresponding $\left(u^{\prime },v^{\prime }\right)$ were computed by mapping (u,v) to **T**. In the same way as calculating the reprojection error for camera calibration, the reprojection error $P_{err}$ of each checkboard corner, as well as the mean reprojection error $MP_{err}$ of the entire checkerboard, can be computed by:(26)$$P_{err}=\sqrt{{\left(u^{\prime \prime }-u^{\prime }\right)}^{2}+{\left(v^{\prime \prime }-v^{\prime }\right)}^{2}}$$(27)$$MP_{err}=\frac{1}{m}\sum_{i=1}^{m}P_{err}$$

The calculation results of $P_{err}$ and $MP_{err}$ for each pose are plotted as a heatmap shown in Figure 11, where the coordinates of each small square correspond to the world coordinates of each corner of the checkboard, and each square’s color represents the value of $P_{err}$ at that position.

From the results, it can be concluded that **T** can achieve high accuracy when the camera is well calibrated. Analyzed from the $MP_{err}$, the $MP_{err}$ of all poses are slightly below the camera calibration’s mean reprojection error of 0.055 pixels. This indicates that the accuracy of the proposed method is closely related to the camera calibration accuracy. Analyzed from the $P_{err}$, the majority of $P_{err}$ are below 0.05 pixels, especially in the central region (6 × 7 = 42 checkboard corners) where no significant outliers are observed, indicating that the method has the best correction accuracy in the center region of the image. Contrasted with this are the four edge regions where the $P_{err}$ of some checkerboard corners are occasionally close to 0.1 pixels. This can be caused by image noise or errors in nonlinear distortion model.

**Exp.2**: Evaluate the residual perspective distortion in the corrected images. The homography matrix $H_{real}^{\prime }$ corresponding to $\left(u^{\prime \prime },v^{\prime \prime }\right)$ and $\left(X_{w},Y_{w}\right)$ was computed by the study’s method. Further, the extrinsic parameters of the measurement plane relative to the corrected imaging plane, i.e., the Euler angles $\alpha^{\prime }$, $\beta^{\prime }$, $\gamma^{\prime }$ and the depth value $t_{3}^{\prime }$, were computed by substituting $K^{\prime }$ and $H_{real}^{\prime }$ to Equations (2) and (15).

The corrected imaging plane constructed in this study is completely parallel to the measurement plane, so the value of $\left|\beta^{\prime }\right|$ and $\left|\gamma^{\prime }\right|$ should be close to zero. At the same time, the α and $t_{3}$ of the actual imaging plane relative to the measurement plane are preserved in the corrected images. So the α error $\left|\alpha_{err}\right|$ and the depth error $\left|D_{err}\right|$, defined as follows, should be close to zero.(28)$$\left|\alpha_{err}\right|=\left|\alpha -\alpha^{\prime }\right|$$(29)$$\left|D_{err}\right|=\left|t_{3}-t_{3}^{\prime }\right|$$

The computed extrinsic parameters of the corrected images are shown in Table 4. The computed evaluation indicators: $\left|\beta^{\prime }\right|$, $\left|\gamma^{\prime }\right|$, $\left|\alpha_{err}\right|$ and $\left|D_{err}\right|$ are shown in Figure 12.

From the results, it can be seen both $\left|\beta^{\prime }\right|$ and $\left|\gamma^{\prime }\right|$ of all poses are less than 0.014°, and especially in some poses, their values are less than 0.005°, which proves that the corrected imaging plane is almost completely parallel to the measurement plane. Hence, the proposed method can effectively correct the perspective distortion. For all poses, $\left|\alpha_{err}\right|$ are less than 0.001° and $\left|D_{err}\right|$ are within 0.016 mm, which proves that the proposed method can effectively preserve the original rotation angle and depth value of the measurement plane relative to the actual imaging plane along the optical axis direction.

**Exp.3**: Compare the measurement accuracy with the existing methods. The pixel equivalents for each of the fifty corrected images were calibrated from Equation (21). The measurement values of R16, L36, H22, D37, D53, and Area were computed by the calibrated pixel equivalents. Then, their corresponding measurement errors are obtained by subtracting the true values from the measurement values.

For comparison purposes, two of the most widely used methods were implemented. One is to measure these parameters by calibrating the homography matrix $H_{real}$ used in Ref. [17] (labeled Miao2020). The other is to measure these parameters through image perspective transformation used in Ref. [22] (labeled Wang2023), in which **T** is calibrated by selecting (u,v) and manually setting their corresponding $\left(u^{\prime },v^{\prime }\right)$ as control point pairs.

Note that the three methods are identical in image processing and parameter calculation. After Canny edge detection to the corrected images, R16 is obtained by fitting the circle using the least square method; L36 and H22 are obtained by calculating the distance between vertices after fitting the lines with the least square method; D37 and D53 are obtained through the fitted lines’ slope. Because the ellipse pattern is viewed as an irregular shape, Area is obtained by counting the number of pixels after binarizing the corrected images using the OTSU method, so Miao2020 cannot be applied to this situation.

To visualize the distribution of the results, the measurement errors derived from the three methods are plotted into standard box plots, as shown in Figure 13.

From the results, the errors of the proposed method are within [−0.03 mm, 0.03 mm] when measuring R16, which is more centralized than the other two methods. In measuring L36 and H22, the errors of the proposed method are within [−0.08 mm, 0.07 mm] and [−0.14 mm, 0.08 mm], respectively, which are more centralized than Miao2020. Although Wang2023 has the best concentration, several outliers appeared. In measuring D37 and D53, the errors of the proposed method are within [0.09°, 0.19°] and [−0.22°, −0.09°], respectively, which are better distributed than Wang2023 and have similar performance to Miao2020. When measuring Area, the accuracy of the proposed method is within [−1.3 mm^2^, 1 mm^2^], which is superior to Wang2023 in terms of measurement accuracy and distribution.

To further evaluate the measurement performance of the three methods, the root mean square errors (RMSEs) and the standard deviations (SDs) of the measurement errors were computed to compare measurement accuracy and stability, respectively. The formulas for RMSE and SD are defined as follows:(30)$$RMSE=\sqrt{\frac{1}{50}\sum_{i=1}^{50}{\left(X_{real}-X_{i}\right)}^{2}}$$(31)$$SD=\sqrt{\frac{1}{50}\sum_{i=1}^{50}{\left(X_{i}-{\bar{X}}_{i}\right)}^{2}}$$
where $X_{real}$ represents the true value, $X_{i}$ represents the measurement value, and ${\bar{X}}_{i}$ represents the mean value of the measurement values.

The RMSE of the three methods is shown in Figure 14, and the SD is shown in Figure 15.

From the results, the RMSE of R16, L36, H22, D37, D53, and Area in the proposed method are within 0.016 mm, 0.052 mm, 0.050 mm, 0.14°, 0.16°, and 0.68 mm^2^, and the SD are within 0.016 mm, 0.043 mm, 0.045 mm, 0.025°, 0.033°, and 0.65 mm^2^, respectively. The proposed method maintains higher accuracy and stability compared to existing methods and even outperforms existing methods in some cases. In particular, when measuring the Area, the RMSE of the proposed method is 32% lower than Wang2023, and the SD is 34% lower than Wang2023. In addition, the proposed method does not need to manually set (u′,v′) as in Wang2023, which avoids the image scaling problem caused by improperly setting the control point pairs.

Overall, the proposed method can effectively improve the accuracy of the pixel equivalent method; thus, it can be applied to realize the high-precision planar measurement under perspective distortion.

Exp. 4: Investigate the impact of camera calibration errors on the proposed method. Pose 1 was selected as the experimental subject, with the camera calibration results from the experiment serving as the ground truth. During the calibration process of **T**, different error amounts ($\Delta f_{x}$, $\Delta f_{y}$, Δs, Δβ, Δγ) were added to parameters $f_{x}$, $f_{y}$, s, β, and γ, respectively, to perform calibration and generate corresponding corrected images. Subsequently, measurements of L36 and H22 were extracted from these corrected images to evaluate the sensitivity of the proposed method to camera calibration errors. The experimental results are illustrated in Figure 16.

According to Equation (21), the increase in errors of $f_{x}$ and $f_{y}$ would reduce M, thereby enlarging the measurement errors of L36 and H22. This is effectively proved by experimental results. Specifically, the error of $f_{x}$ mainly affects the measurement result of L36, and the error of $f_{y}$ mainly affects the measurement result of H22. The error in s has a relatively minor impact on measurement results, but its influence intensifies with increasing pixel length. The error in β primarily causes scaling along the x-direction, leading to reduced pixel length of L36 and consequently larger measurement errors. Similarly, the error in γ mainly induces scaling along the y-direction, resulting in shortened pixel length of H22 and amplified measurement errors. Overall, the proposed method can achieve satisfactory measurement performance when camera calibration errors remain within acceptable limits.

Additionally, $\beta^{\prime }$ and $\gamma^{\prime }$ were further calculated from the corrected images to assess the planar tilt residuals caused by camera calibration errors. The experimental results are presented in Figure 17.

From the results, it can be seen that among the intrinsic parameters, errors in $f_{x}$ primarily induce $\left|\beta^{\prime }\right|$, errors in $f_{y}$ mainly cause $\left|\gamma^{\prime }\right|$, and errors in s simultaneously contribute to both $\left|\beta^{\prime }\right|$ and $\left|\gamma^{\prime }\right|$. But relatively speaking, the impact of intrinsic parameter errors on tilt residuals is relatively small. In contrast, within the extrinsic parameters, $\left|\beta^{\prime }\right|$ is almost equivalent to Δβ and $\left|\gamma^{\prime }\right|$ nearly equals Δγ. Thus, it can be concluded that when the camera is well calibrated, extrinsic parameter errors become the dominant factor causing tilt residuals.

## 5. Conclusions

In monocular vision measurement, a barrier to implementation is the perspective distortion caused by manufacturing errors in the imaging chip and non-parallelism between the measurement plane and its image, which makes it challenging to improve the accuracy of pixel equivalent and measurement results. To address this issue, the paper proposed a perspective distortion correction method for planar imaging based on homography mapping.

This method overcomes the limitations of traditional approaches that require the manual setting of ideal points for calibrating the perspective transformation model. Instead, it achieves calibration solely through the homography relationship between planes. Furthermore, the proposed method integrates the perspective transformation model with the nonlinear distortion model, enabling simultaneous correction of both perspective and nonlinear distortions through a single image interpolation process.

In experiments, the proposed method demonstrated high accuracy with a mean reprojection error of less than 0.05 pixels. It effectively corrects distortions while preserving the original rotation angle and depth value of the measurement plane relative to the actual imaging plane along the optical axis. In measuring the radius, length, angle, and area of the designed pattern, the RMSE of the proposed method are within 0.016 mm, 0.052 mm, 0.16°, and 0.68 mm^2^, with the SD of 0.016 mm, 0.045 mm, 0.033°, and 0.65 mm^2^, respectively. Compared to the existing methods, the proposed method exhibited lower RMSE and SD (specifically 32% and 34% lower when measuring area, respectively), proving the proposed method has higher accuracy and stability. The proposed method can effectively improve the accuracy of the pixel equivalent method, thus realizing high-precision planar measurement under perspective distortion.

Based on current research, future work will focus on two aspects: firstly, integrating the proposed method with downstream edge detection algorithms to achieve measurement of complex targets in industrial scenarios, and secondly, optimizing algorithm design to meet the requirements of large field-of-view and high real-time performance in industrial visual measurement systems.

## Figures and Tables

**Figure 1 sensors-25-01891-f001:**
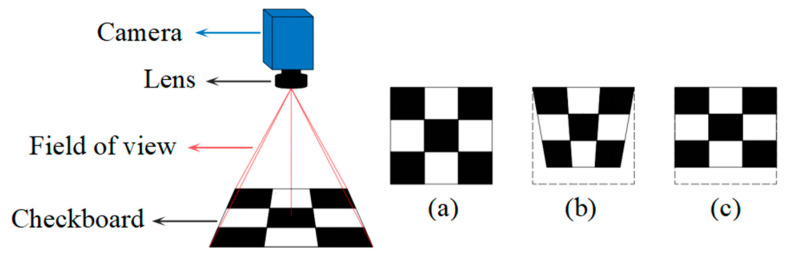
Perspective distortion under different lenses (The gray dash line represents imaging field of view): (**a**) Actual pattern; (**b**) imaging under ordinary lens; (**c**) imaging under telecentric lens.

**Figure 2 sensors-25-01891-f002:**
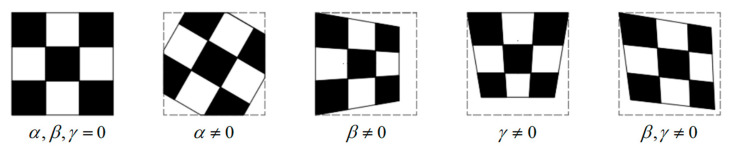
Impact of the three Euler angles on plane imaging (The gray dash line represents imaging field of view).

**Figure 3 sensors-25-01891-f003:**
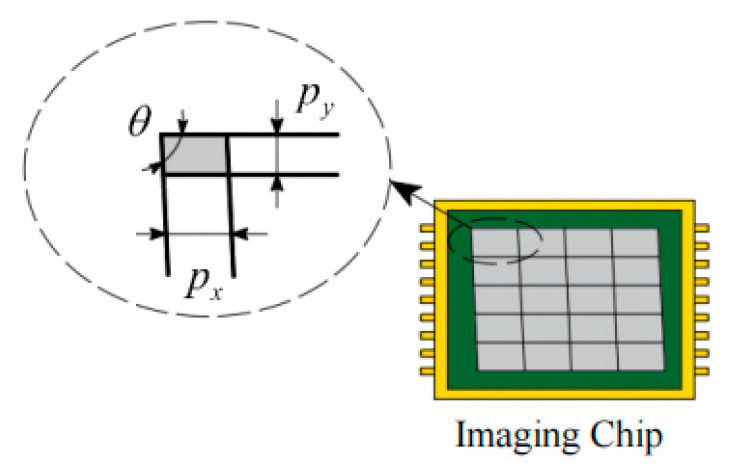
Perspective distortion caused by intrinsic parameters.

**Figure 4 sensors-25-01891-f004:**
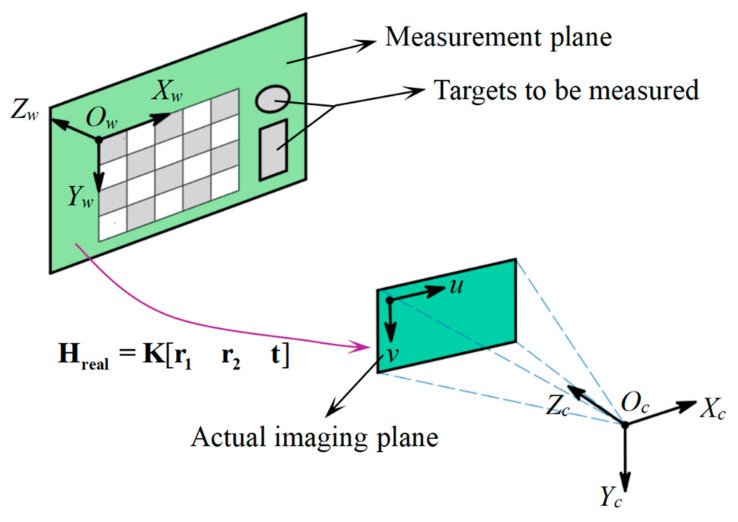
Calibration of the actual homography matrix (The purple line represents the transformation relationship, and the blue dash line represents the field of view of the actual camera).

**Figure 5 sensors-25-01891-f005:**
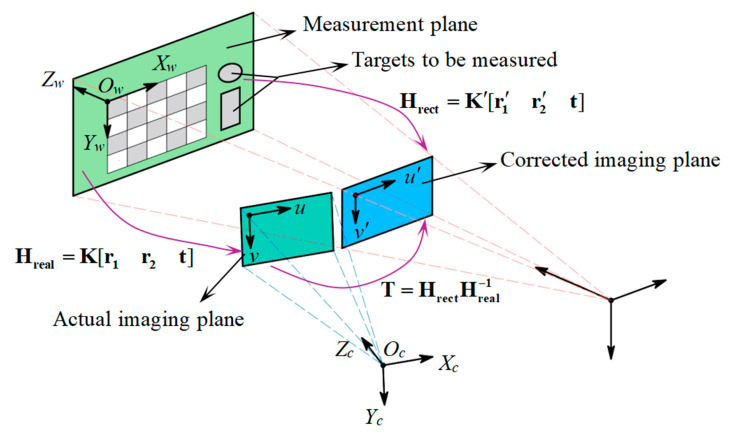
Homography relationship between planes (The purple line represents the transformation relationship, the blue dash line represents the field of view of the actual camera, and the red dash line represents the field of view of the ideal camera).

**Figure 6 sensors-25-01891-f006:**
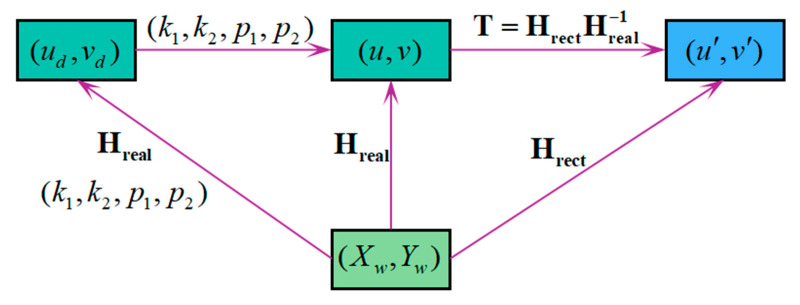
Relationship between coordinate points (The purple line represents the transformation relationship).

**Figure 7 sensors-25-01891-f007:**
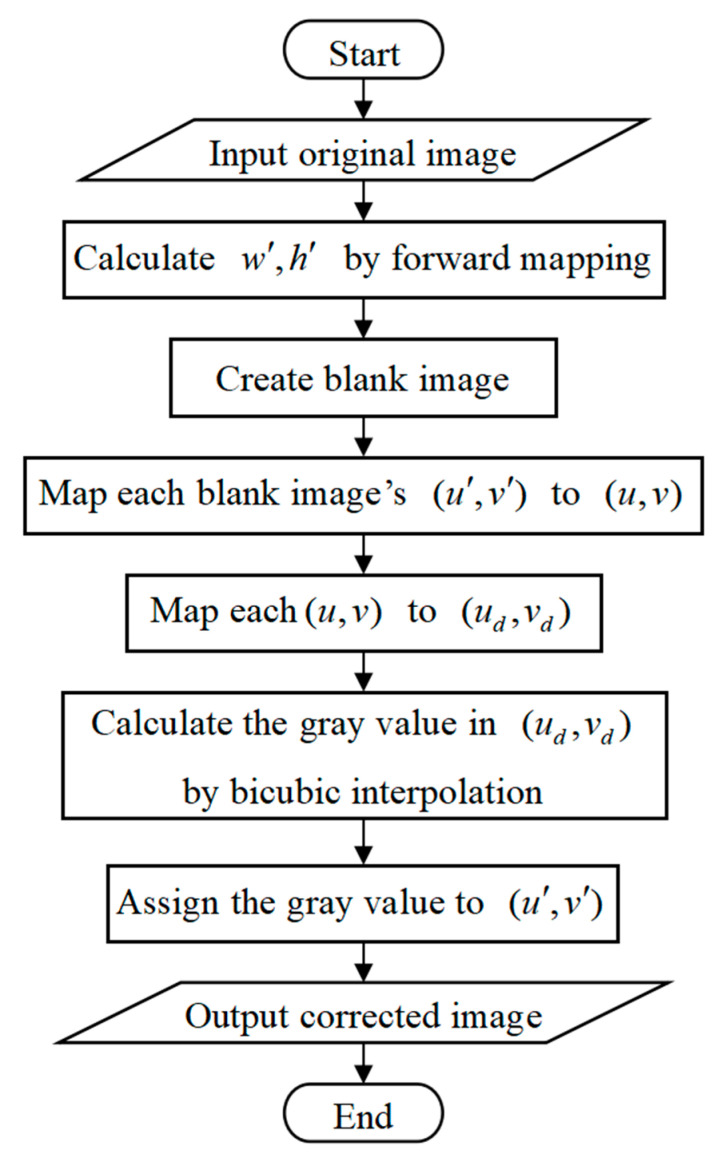
Process of distortion correction.

**Figure 8 sensors-25-01891-f008:**
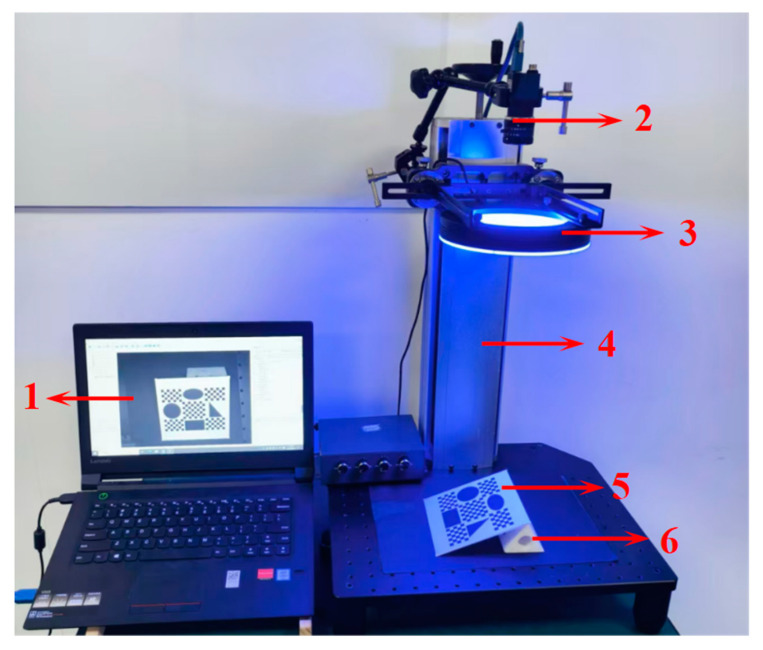
The visual measurement platform: (1) PC; (2) industrial camera; (3) ring light; (4) machine vision platform; (5) test board; (6) adjustment block.

**Figure 9 sensors-25-01891-f009:**
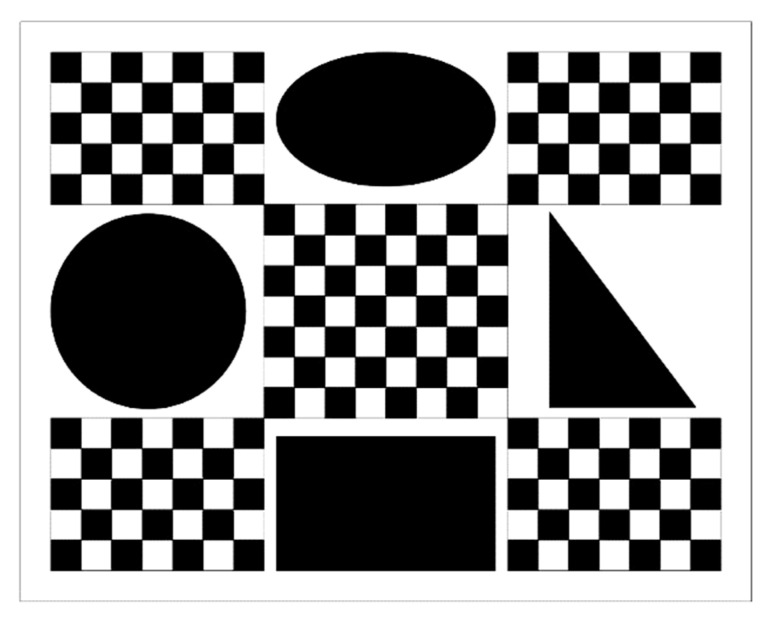
Pattern of the test board.

**Figure 10 sensors-25-01891-f010:**
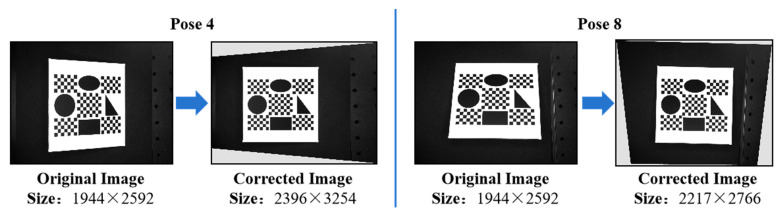
Comparison before and after correction.

**Figure 11 sensors-25-01891-f011:**
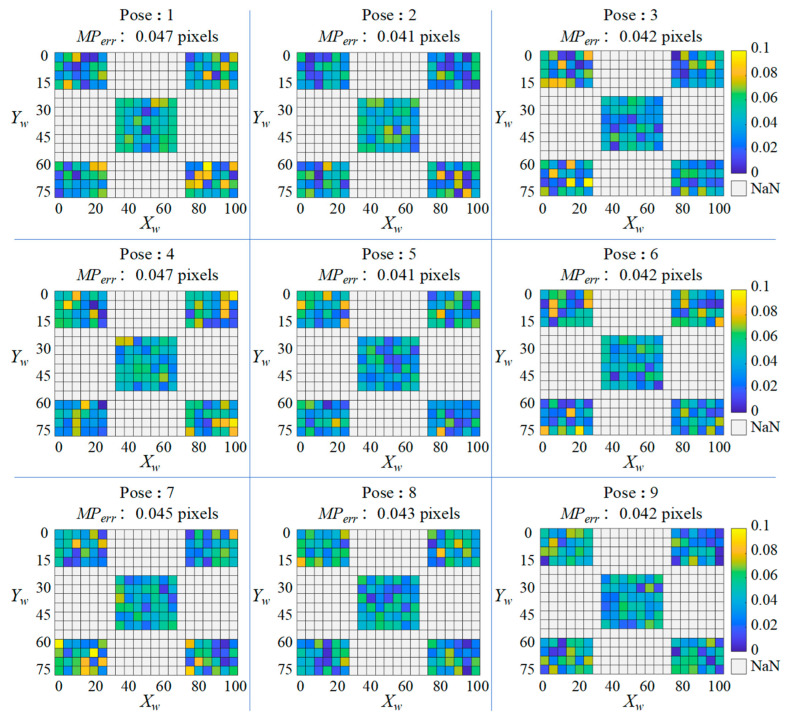
The calculation results of $P_{err}$ and $MP_{err}$ for each pose.

**Figure 12 sensors-25-01891-f012:**
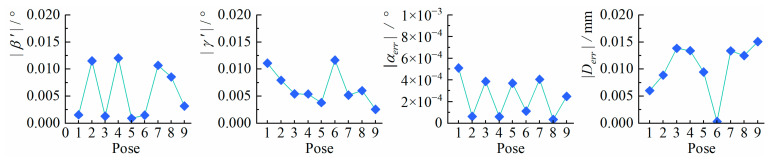
The calculation results of $\left|\beta^{\prime }\right|$, $\left|\gamma^{\prime }\right|$, $\left|\alpha_{err}\right|$ and $\left|D_{err}\right|$ for each pose.

**Figure 13 sensors-25-01891-f013:**
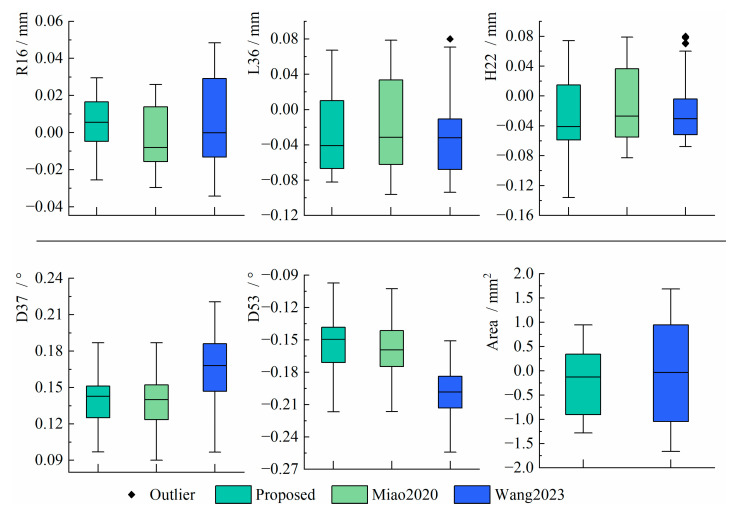
The distribution of measurement errors [17,22].

**Figure 14 sensors-25-01891-f014:**
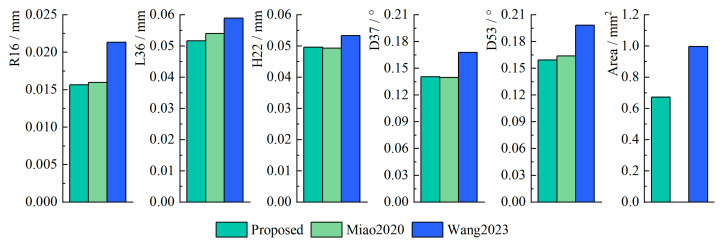
The RMSE of the three methods [17,22].

**Figure 15 sensors-25-01891-f015:**
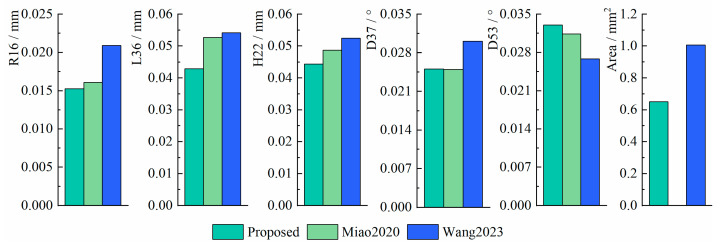
The SD of the three methods [17,22].

**Figure 16 sensors-25-01891-f016:**
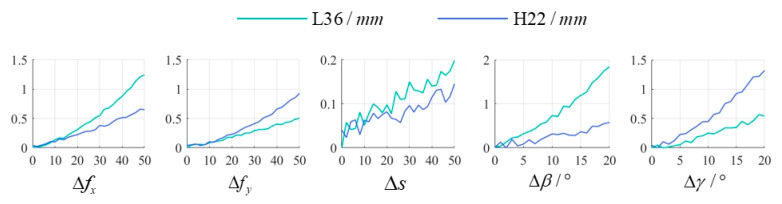
The sensitivity of the proposed method to camera calibration errors.

**Figure 17 sensors-25-01891-f017:**
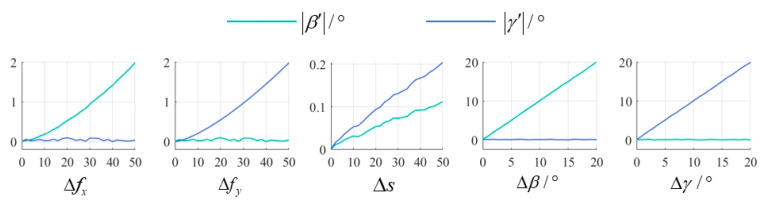
The planar tilt residuals caused by camera calibration errors.

**Table 1 sensors-25-01891-t001:** Physical parameters of the industrial camera.

Parameters Names	Parameters Values
Pixel resolution	2592 × 1944
Pixel size	2.2 μm × 2.2 μm
Size of imaging chip	2592 × 2.2 = 5.702 mm
	1944 × 2.2 = 4.276 mm
Focal length	12 mm
Focus distance	520 mm
Image distance	1/(1/12 − 1/520) = 12.283 mm
Visual field	520 × 5.702/12.283 = 241.394 mm
	520 × 4.276/12.283 = 181.024 mm
Pixel precision	520 × 2.2/12.283 = 93.137 μm/pixel

**Table 2 sensors-25-01891-t002:** Calibration results of the camera intrinsic parameters.

Parameters Names	Parameters Values
$\left(u_{0},v_{0}\right)$	(1262.928, 960.322)
$f_{x}$	5497.031
$f_{y}$	5497.245
s	0.041
$k_{1}$	−0.077
$k_{2}$	0.305
$p_{1}$	0.00047
$p_{2}$	−0.00011

**Table 3 sensors-25-01891-t003:** Extrinsic parameters of the original images.

Pose	*α*/°	*β*/°	*γ*/°	*t*_3_/mm
1	−0.198	−16.987	−0.316	521.447
2	−0.897	−8.157	−0.350	523.331
3	−0.579	13.439	−0.240	494.681
4	−0.113	29.971	−0.132	464.244
5	1.287	−1.827	−33.269	470.868
6	−0.238	−1.384	−16.685	495.230
7	−0.364	−1.990	6.645	522.048
8	−1.330	−1.699	30.047	512.587
9	4.653	7.897	−8.376	491.392

**Table 4 sensors-25-01891-t004:** Extrinsic parameters of the corrected images.

Pose	*α*′/°	*β*′/°	*γ*′/°	$t_{3}^{\prime }/mm$
1	−0.199	0.002	−0.011	521.453
2	−0.897	−0.012	−0.008	523.340
3	−0.579	0.001	0.005	494.695
4	−0.113	0.012	−0.005	464.258
5	1.287	−0.001	−0.004	470.858
6	−0.238	0.001	−0.011	495.230
7	−0.364	−0.010	−0.005	522.061
8	−1.330	−0.008	0.006	512.599
9	4.653	−0.018	−0.001	491.407

## Data Availability

Dataset available on request from the authors.

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
