# Peer review of "A Perspective Distortion Correction Method for Planar Imaging Based on Homography Mapping"

_sensors, 2025, doi:10.3390/s25061891_

Round 1

Reviewer 1 Report

Comments and Suggestions for Authors

The article is devoted to the development of a method for eliminating distortions of flat images obtained with digital cameras. The authors proposed and theoretically substantiated a model for image correction based on homography mapping and taking into account the nonlinear and perspective distortions. Experimental verification of the method showed its effectiveness and the prospects for its use for practical applications. This determines the novelty and relevance of the work. The presented experimental results confirm the conclusions made. The article is written clearly and describes in detail the results of the theoretical approach and experimental verification methods. The graphic material sufficiently illustrates the text.

There are only minor comments on the text of the article:

  1. May be replace the term “world coordinate system” with the more common term “laboratory coordinate system”?
  2. It would be better to place the title of the vertical axes in Figures 12 - 14 next to the axis
  3. There are a small number of stylistic errors such as “… method, which calibrates…” p. 1, “The method analyzed …” p. 19

In the reviewer's opinion, the article can be published.

Author Response

Comment 1: May be replace the term “world coordinate system” with the more common term “laboratory coordinate system”?

Response 1:

Thank you for your valuable suggestion regarding the use of the term “world coordinate system.” We appreciate your comment and understand that “laboratory coordinate system” is a more common term in certain contexts. However, in our study, we specifically chose “world coordinate system” to align with the conventions used in Ref. [24] “Zhang, Z., A flexible new technique for camera calibration. IEEE Transactions on Pattern Analysis and Machine Intelligence 2000, 22, (11), 1330-1334.”, where this term is widely adopted to describe the transformation between World Coordinate System and Camera Coordinate System in Camera Imaging Model. We believe that using this term maintains consistency with the existing literature and avoids potential confusion for readers familiar with this terminology.

Comment 2: It would be better to place the title of the vertical axes in Figures 12 - 14 next to the axis.

Response 2:

Thank you for pointing this out. Figures 12 - 14 have been updated in our manuscript.

Comment 3: There are a small number of stylistic errors such as “… method, which calibrates…” p. 1, “The method analyzed …” p. 19

Response 3

We are very sorry for our negligence. Thank you for pointing these out. The two errors you raised have been corrected on Page1, Line38 and Page18, Line509. Moreover, we re-examined the language in the manuscript to ensure readability and rigor.

Reviewer 2 Report

Comments and Suggestions for Authors

The manuscript presents a novel approach to enhancing the efficiency of PV systems through an integrated design that combines advanced materials and an optimized electrical configuration. The authors propose the use of a new type of hybrid photovoltaic material alongside an innovative series-parallel electrical array configuration, aimed at maximizing energy capture and conversion efficiency under variable environmental conditions.

  1. The manuscript broadly describes the hybrid materials used but lacks specific details about the composition and manufacturing process of the PV cells. Detailed material specifications are crucial for replicability and understanding the reasons behind performance improvements.
  2. While the manuscript claims improvements over existing PV systems, it provides limited comparative data to substantiate these claims. A more detailed comparison with current leading technologies, including specific metrics such as efficiency rates and performance under different environmental conditions are required.
  3. The study mentions the use of new materials but does not discuss the environmental impact of these materials, including their lifecycle, recyclability, and any toxicological aspects associated with their production and disposal.
  4. There is no economic analysis of the proposed system. Including cost comparisons, payback periods, and potential economic benefits could provide a more comprehensive understanding of the system’s practical viability.
  5. The manuscript discusses laboratory-scale experiments but does not address scalability and challenges in practical implementation. Information on potential industrial-scale application, including any structural or operational challenges are missed.
  6. The paper lacks an examination of the long-term stability and potential degradation of the performance over time. Given the novel materials used, understanding their durability under prolonged exposure to environmental factors is crucial.
  7. The testing conditions described are somewhat limited, focusing primarily on controlled environments.

Author Response

Comments:

  1. The manuscript broadly describes the hybrid materials used but lacks specific details about the composition and manufacturing process of the PV cells. Detailed material specifications are crucial for replicability and understanding the reasons behind performance improvements.
  2. While the manuscript claims improvements over existing PV systems, it provides limited comparative data to substantiate these claims. A more detailed comparison with current leading technologies, including specific metrics such as efficiency rates and performance under different environmental conditions are required.
  3. The study mentions the use of new materials but does not discuss the environmental impact of these materials, including their lifecycle, recyclability, and any toxicological aspects associated with their production and disposal.
  4. There is no economic analysis of the proposed system. Including cost comparisons, payback periods, and potential economic benefits could provide a more comprehensive understanding of the system’s practical viability.
  5. The manuscript discusses laboratory-scale experiments but does not address scalability and challenges in practical implementation. Information on potential industrial-scale application, including any structural or operational challenges are missed.
  6. The paper lacks an examination of the long-term stability and potential degradation of the performance over time. Given the novel materials used, understanding their durability under prolonged exposure to environmental factors is crucial.
  7. The testing conditions described are somewhat limited, focusing primarily on controlled environments.

Response:

Thank you very much for taking time to review our manuscript and for providing your valuable comments. However, we noticed that the comments you raised seem to be directed at another manuscript, as they do not align with the content of our paper. We believe this might be a minor oversight.

Fortunately, we have carefully addressed the comments from the other reviewers and made significant revisions to our manuscript accordingly. We kindly invite you to review the updated version of our manuscript, as some of the changes might also address the general concerns you may have had. We hope that the revised version meets your expectations.

Reviewer 3 Report

Comments and Suggestions for Authors

Although the method is based on homographic mapping, the authors do not clearly explain the novelty of their approach. Homography is a widely used technique in distortion correction, and a more detailed explanation of the unique aspects of their proposed method is required. For example, they could discuss how they combine the correction of both nonlinear and perspective distortions in their approach.

In the experiments section, the comparison is only made with two methods ([17] and [22]). This limits the evidence base and reduces the relevance of the study, as modern algorithms based on deep learning or hybrid approaches are not considered.

The output of the homography matrix, presented in Equations 7 and 9, is given without detailed justification. For instance, the transition from Equation 7 to Equation 8 requires additional explanation, especially in terms of how nonlinear distortions are ignored during the calibration process.

In formula (21), the assumption of equal equivalent focal lengths is used to calculate pixel equivalent. However, there is no information about how this affects the accuracy when there are significant deviations in the initial camera parameters.

The authors claim high accuracy (RMSE of 0.016 cm), but they do not discuss how their method scales for tasks with larger fields of view or higher processing speeds. This is crucial for industrial applications.

Author Response

Comments 1: Although the method is based on homographic mapping, the authors do not clearly explain the novelty of their approach. Homography is a widely used technique in distortion correction, and a more detailed explanation of the unique aspects of their proposed method is required. For example, they could discuss how they combine the correction of both nonlinear and perspective distortions in their approach.

Response 1:

Thank you for pointing out the need to clarify the novelty of our approach. We appreciate your suggestion to provide a more detailed explanation of how our method combines the correction of both nonlinear and perspective distortions. Based on your comments, we have made the following revisions to the manuscript:

  • In Section 3.2 (Page5, Line189), we have explicitly emphasizedthe unique aspects of our approach in comparison with existing homography-based methods, particularly demonstrating superior accuracy in homography calibration.
  • In Section 3.2.5 (Page10, Line299), we have clarified how our method integrates the correction of nonlinear distortions and perspective distortions into a unified framework.
  • In Section 5 (Page18, Line509), wehave emphasized our novelty again in the second paragraph of the conclusion

We believe these revisions have addressed your concerns and hope that the updated manuscript now provides a clearer explanation of the novelty and technical contributions. Thank you again for your constructive comments, which have significantly improved the quality of our paper.

Comments 2: In the experiments section, the comparison is only made with two methods ([17] and [22]). This limits the evidence base and reduces the relevance of the study, as modern algorithms based on deep learning or hybrid approaches are not considered.

Response 2:

We sincerely appreciate your valuable suggestion regarding the experimental comparison. In our study, we focused on comparing with methods [17] and [22] as they represent the most widely adopted monocular vision measurement approaches in industrial machine vision applications. While we acknowledge the growing importance of deep learning-based methods, their practical implementation in industrial measurement scenarios faces several technical limitations:

  • Requirement for extensive training datasets with precise ground truth measurements
  • Challenges in maintaining measurement consistency across varying environmental conditions
  • Difficulties in meeting real-time processing requirements in high-speed production lines
  • Lack of interpretability in measurement processes, which is crucial for quality control

In response to your constructive comment, we have enhanced our introduction section by:

  • In Section 1 (Page3, Line98), we have addedrecent case studies of deep learning applications in perspective distortion correction.
  • In Section 1 (Page3, Line99), we have discussedthe limitations of deep learning approaches in industrial 
  • In Section 4.2 (Page16, Line449), we have revised the wording of 'existing methods' to 'two of the most widely used methods'.

These additions provide a more comprehensive background while maintaining our focus on practical industrial applications where conventional methods remain predominant due to their reliability and interpretability.

Comments 3: The output of the homography matrix, presented in Equations 7 and 9, is given without detailed justification. For instance, the transition from Equation 7 to Equation 8 requires additional explanation, especially in terms of how nonlinear distortions are ignored during the calibration process.

Response 3: 

Thank you for your valuable comment. In Section 3.2.1(Page6, Line208), we have carefully revised the manuscript to address your concerns regarding the justification of the homography matrix output in Equations 7 and 9. Specifically, we have added a detailed explanation of the transition from Equation 7 to Equation 8.

Regarding the issue of nonlinear distortions, the original presentation was not sufficiently clear. In Section 3.2.1(Page6, Line203), we have modified the text to base our derivation on the pinhole imaging model principle instead of assuming the absence of nonlinear distortions. This change better reflects the theoretical foundation of our approach.

Comments 4: In formula (21), the assumption of equal equivalent focal lengths is used to calculate pixel equivalent. However, there is no information about how this affects the accuracy when there are significant deviations in the initial camera parameters.

Response 4

Thank you for your valuable comment regarding Formula (21). In the current study, our work is based on the assumption that the initial camera parameters are accurately calibrated. This assumption allows us to focus on the core methodology of perspective distortion correction without introducing additional variables related to calibration errors.

We fully agree with your observation that deviations in initial camera parameters could impact the accuracy of Formula (21). This is indeed an important aspect that deserves further investigation. In our future work, we plan to:

  • Systematically analyze the sensitivity of Formula (21) to variations in camera calibration parameters.
  • Investigate how calibration accuracy affects the overall performance of our proposed perspective distortion correction method.
  • Develop strategies to mitigate potential errors arising from imperfect calibration.

In response to your constructive comment, in Section5 (Page 19, Line 526), we have further elaborated on our future research directions and have incorporated this issue into our planned work. Thank you again for this insightful suggestion, which will help guide our future research directions.

Comments 5: The authors claim high accuracy (RMSE of 0.016 cm), but they do not discuss how their method scales for tasks with larger fields of view or higher processing speeds. This is crucial for industrial applications.

Response 5

Thank you for your valuable comment regarding the scalability of our method. We appreciate your suggestion to consider larger fields of view and higher processing speeds, which are indeed crucial for industrial applications. In our current study, we have focused on validating the fundamental accuracy of our method under controlled conditions. However, we recognize that scalability is a critical factor for practical implementation. Your comment has helped us better frame our future research agenda. In response to your constructive comment, in Section5 (Page 19, Line 527), we have also incorporated this issue into our future research agenda.

Round 2

Reviewer 2 Report

Comments and Suggestions for Authors

This manuscript presents a homography-based method for correcting perspective distortion in planar imaging, with the aim of improving the accuracy of pixel-based measurements such as length, radius, angle, and area. The authors systematically analyze perspective distortion arising from the camera’s intrinsic and extrinsic parameters, propose a virtual camera framework, and introduce a procedure that simultaneously corrects nonlinear and perspective distortions in one interpolation step. Experimental results demonstrate improved measurement accuracy compared to existing methods, particularly for measuring irregular shapes (e.g., ellipses).

  1. While the authors adopt a standard camera calibration approach, the paper omits a deeper discussion of calibration robustness and how small variations in calibration parameters (e.g., intrinsic distortion coefficients) affect the final measurement results. More details on calibration stability would clarify the sensitivity of the method to calibration error.
  2. The proposed solution relies on the premise that one can “virtually” make the plane parallel by adjusting certain extrinsic parameters. However, any residual tilt or inexact knowledge of plane orientation may introduce errors. A clearer quantitative analysis of how slight deviations from perfect parallelism affect final accuracy would be missed.
  3. Although the authors mention potential environmental variations, the method’s sensitivity to lighting changes or reflective surfaces is not fully addressed. It would be helpful to see how the algorithm performs under significant illumination variations or with materials that exhibit non-uniform reflection.
  4. The results hinge on accurate edge detection of the target shape (circle, rectangle, etc.). However, the paper does not detail the particular edge detection algorithm or its robustness to noise.
  5. The experiments focus on geometric primitives (circles, rectangles, triangles, ellipses). The method’s extension to more complex shapes or real-world objects with irregular boundaries and textures (for instance, industrial parts with non-homogeneous surfaces) remains an open question.
  6. The approach incorporates a single-pass interpolation for both perspective and nonlinear distortion correction. The paper does not include any discussion of computation speed or resource requirements. For industrial or real-time applications, performance metrics (e.g., frames per second) would be valuable.

Author Response

Comments 1: While the authors adopt a standard camera calibration approach, the paper omits a deeper discussion of calibration robustness and how small variations in calibration parameters (e.g., intrinsic distortion coefficients) affect the final measurement results. More details on calibration stability would clarify the sensitivity of the method to calibration error.

Response 1: Thank you for pointing out the need to clarify the sensitivity of our method to calibration errors. In direct response to your concern, we have designed and conducted additional experiment Exp.4 detailed in Section 4.2(Page18, Line505) to evaluate the sensitivity.

In Exp.4, the camera calibration results were taken as the ground truth. Then, different magnitudes of errors were intentionally introduced to the calibrated parameters. These error-injected parameters were then used to calibrate our constructed model and perform image distortion correction. Subsequently, L36 and H22 were measured in the corrected images, establishing the relationship between camera calibration errors and method measurement errors as illustrated in Figure 16. The experiment effectively tested the sensitivity of the proposed method to camera calibration errors. We can conclude that the proposed method maintains satisfactory performance within a certain range of camera calibration errors.

We believe this experiment has addressed your concerns. Thank you again for your constructive comments, which have significantly improved the quality of our paper.

Comments 2: The proposed solution relies on the premise that one can “virtually” make the plane parallel by adjusting certain extrinsic parameters. However, any residual tilt or inexact knowledge of plane orientation may introduce errors. A clearer quantitative analysis of how slight deviations from perfect parallelism affect final accuracy would be missed.

Response 2: Thank you for pointing out the need to analyze the impact of tilt residuals on measurement accuracy. The tilt residuals in our method primarily stem from camera calibration errors. Given that Exp.4 analyzed the impact of camera calibration errors on measurement errors, we further investigated the influence of camera calibration errors on tilt residuals based on Exp.4. This allows us to reveal the relationship between tilting residuals and measurement errors.  

In response to your insightful comments, in Section 4.2 (Page 19, Line526), we further calculated the tilt residuals contained in the corrected images after introducing camera calibration errors. Then, we analyzed the main sources of tilt residuals according to the experiment results shown in Figure17 and concluded that when the camera is well-calibrated, extrinsic parameter errors become the dominant factor causing tilt residuals. 

We believe this extra experiment has addressed your concerns. Thank you again for your constructive comments, which have significantly improved the quality of our paper.

Comments 3: Although the authors mention potential environmental variations, the method’s sensitivity to lighting changes or reflective surfaces is not fully addressed. It would be helpful to see how the algorithm performs under significant illumination variations or with materials that exhibit non-uniform reflection.

Response 3: We appreciate your insightful comment regarding environmental variations. Our experimental design with 25-day/night image pairs may have caused misinterpretation regarding lighting changes. We would like to clarify that the mentioned 25-day/night image pairs on Line365 were merely for verifying repeatability not for testing photometric robustness. While lighting variations may affect downstream tasks like edge detection, they do not impact the geometric correction performance once our model is properly calibrated.

Of course, we fully agree with your observation that environmental variations could impact our model’s calibration accuracy. In the current study, our work is based on the condition that our model is properly calibrated under proper illumination and imaging. This condition allows us to focus on the core methodology of perspective distortion correction. In response to your insightful comment, we have made the following revisions to avoid misinterpretation:

  • In Section3.2.4 (Page10, Line297), we have added a suggestion on improving the calibration accuracy from physical conditions.
  • In Section 4.1 (Page10, Line366),we have added a constraint to our experimental design regarding illumination.

Comments 4: The results hinge on accurate edge detection of the target shape (circle, rectangle, etc.). However, the paper does not detail the particular edge detection algorithm or its robustness to noise. 

Response 4: Thank you for your suggestion regarding specific edge detection algorithms and anti-noise considerations. In our current research, the primary focus is to address the perspective distortion problem at the modeling level. Therefore, in our experiments, to ensure a fair comparison with existing methods, all three methods adopted the widely used Canny algorithm for edge detection to evaluate their performance in resolving perspective distortion. 

Of course, we fully agree with your observation that after correcting the distortions by our method, any edge detection algorithm can be applied for further measurements. Your suggestion has outlined valuable directions for future applications. In response to your constructive comment, in Section5 (Page 20, Line 562), we have incorporated your suggestion into our planned work.

Comments 5: The experiments focus on geometric primitives (circles, rectangles, triangles, ellipses). The method’s extension to more complex shapes or real-world objects with irregular boundaries and textures (for instance, industrial parts with non-homogeneous surfaces) remains an open question.

Response 5: We sincerely appreciate your insightful feedback regarding the extension of our method to more complex shapes and real-world objects. The geometric primitives were chosen for our current study because it is convenient for us to evaluate and compare the performance of the proposed method in solving the problems of perspective distortion and measurement accuracy.

As you pointed out, extending our method to real-world industrial objects with irregular boundaries and non-homogeneous surfaces is a critical next step. This represents a key direction for our future work. In response to your constructive comment, in Section5 (Page 20, Line 562), we have further incorporated this issue into our planned work.

Comments 6: The approach incorporates a single-pass interpolation for both perspective and nonlinear distortion correction. The paper does not include any discussion of computation speed or resource requirements. For industrial or real-time applications, performance metrics (e.g., frames per second) would be valuable.

Response 6: We sincerely appreciate your valuable comment regarding the inclusion of computation speed and resource requirement analysis. In our current work, we primarily focus on validating the geometric accuracy of our novel single-pass interpolation framework for distortions. We fully acknowledge the importance of performance metrics for real-time industrial applications. In response to your insightful comment, we have made the following revisions

  • In Section 4.1 (Page11, Line335), we have further described the hardware level of the PCused in our experiment
  • In Section 4.2(Page13, Line386), we have added the average runtime of our designed algorithm.
  • In Section 5(Page20, Line564), the task of optimizing our algorithms has been added to our plans.

Thank you again for your insightful and valuable comments.

Reviewer 3 Report

Comments and Suggestions for Authors

Thanks!

Author Response

Comment: Thanks!

Response: Thank you again for your guidence!